# Sources, Selection, and Microenvironmental Preconditioning of Cells for Urethral Tissue Engineering

**DOI:** 10.3390/ijms232214074

**Published:** 2022-11-15

**Authors:** Zongzhe Xuan, Vladimir Zachar, Cristian Pablo Pennisi

**Affiliations:** Regenerative Medicine Group, Department of Health Science and Technology, Aalborg University, 9220 Aalborg, Denmark

**Keywords:** urology, urethra, mesenchymal stem cells, smooth muscle cells, keratinocytes, urine-derived stem cells

## Abstract

Urethral stricture is a common urinary tract disorder in men that can be caused by iatrogenic causes, trauma, inflammation, or infection and often requires reconstructive surgery. The current therapeutic approach for complex urethral strictures usually involves reconstruction with autologous tissue from the oral mucosa. With the goal of overcoming the lack of sufficient autologous tissue and donor site morbidity, research over the past two decades has focused on cell-based tissue-engineered substitutes. While the main focus has been on autologous cells from the penile tissue, bladder, and oral cavity, stem cells from sources such as adipose tissue and urine are competing candidates for future urethral regeneration due to their ease of collection, high proliferative capacity, maturation potential, and paracrine function. This review addresses the sources, advantages, and limitations of cells for tissue engineering in the urethra and discusses recent approaches to improve cell survival, growth, and differentiation by mimicking the mechanical and biophysical properties of the extracellular environment.

## 1. Introduction

Urethral stricture is a urinary tract disorder with an estimated incidence of 0.6% in men, which can be the result of a variety of factors, including trauma, infection, and iatrogenic causes [1]. Common to all of these conditions is that the formation of fibrous scar tissue narrowing the urethral lumen can lead to symptoms such as incomplete bladder emptying, increased frequency of urination, and difficulty or pain with urination, all of which negatively impact the patient’s quality of life. If left untreated, strictures can lead to complications such as incontinence, urinary tract infections, or even kidney failure [2]. In recent decades, the prevalence of strictures has remained at historically high levels. According to statistics, the total cost of urethral strictures in the United States is more than $6000 per affected person, with annual medical costs of nearly $200 million in 2009, and spending is currently much higher [3]. Because urethral stricture imposes a significant social and economic burden on individuals, families, and the health care system, the medical community has turned its focus to finding new therapeutic solutions.

The current gold standard for the treatment of complex urethral strictures is urethroplasty with oral mucosa [4]. Oral mucosa has many properties that make it particularly suitable for urethral reconstruction, as it adapts well to the moist environment of the urethra, has high mechanical strength, and can be rapidly vascularized by blood supply from the wound bed. It is also easy and quick to harvest, has a hidden suture line, and can lead to long-term success rates of 80–85% [5]. However, only a limited amount of oral mucosa is available for harvesting, and complications such as bleeding at the donor site, infection, pain, parotid duct injury, graft contracture, and numbness may occur [6,7].

Tissue engineering is a promising approach to compensate for the lack of autologous tissue and can avoid the complications associated with harvesting of the grafts [8,9,10]. Tissue engineering approaches often require biodegradable scaffolds that can serve as guidance and structural platforms for progenitor and stem cells to regenerate tissue [11]. The simplest tissue engineering strategy for urethral reconstruction involves cell-free natural or synthetic scaffolds [12]. Host cells infiltrate the scaffolds, which are remodeled and eventually replaced by the target tissue [13]. However, the success of acellular grafts depends on a healthy urethral bed, adequate vascularity, and the absence of spongy fibrosis. Otherwise, graft shrinkage, inadequate tissue regeneration, and uncontrolled fibrotic tissue formation may occur [14]. Since the underlying pathologic process in stricture disease is ischemic spongy fibrosis, the quality of the wound bed may be compromised [15]. Furthermore, this simple strategy can only be used as a backup option in patients with short to moderate urethral defects, as clinical data show that failures are common in patients with long strictures (>4 cm) [16]. For the treatment of complex strictures, cell-based tissue-engineered constructs have been investigated. A systematic review has shown that cell-based grafts have a 5.7-fold higher long-term success rate than unseeded grafts [9]. Notably, cell-loaded grafts have been shown to reduce the incidence of strictures, fistulas, and infections [14]. The most likely explanation is that cellularization may promote vascularization and urothelial barrier formation, both of which can effectively reduce local inflammation and fibrosis caused by urine leakage [11]. Therefore, the selection of the appropriate cell type in conjunction with a supportive scaffold is a crucial step in tissue engineering for urethral reconstruction [17]. While the use of cell-loaded scaffolds appears promising for the repair of complex urethral strictures, the optimal conditions for cell maturation and differentiation are still not well understood. The aim of this review is to describe the different cell types that can be used for urethral reconstruction and the techniques and conditions that can be used to optimize the proliferation, engraftment, and functional maturation of these cells. The review includes cells with proven efficacy in clinical trials as well as cell types that have only recently been studied in vitro and in preclinical testing but represent a potential source for clinical use.

## 2. Overview of the Structure and Function of the Male Urethra

The urethra’s primary function is to transport urine from the bladder for excretion from the body and, in males, to serve as a conduit for sperm [18]. The bulbar and penile urethras are found in the anterior portion of the urethra, which is also known as the spongy urethra because it is surrounded by the corpus spongiosum. The urethral epithelium and lamina propria (mucosa), submucosa, and muscle layer are histologically present in the spongy urethra. The epithelial layer of the urethra acts as a highly impermeable barrier to the toxic substances contained in urine, and its histological profile varies from segment to segment. While the mucosa near the bladder neck consists of transitional epithelium, the anterior urethra is lined by pseudostratified columnar epithelium, and the external orifice (meatus urethralis) is lined by stratified squamous epithelium [18]. The bulbar urethra is small and fixed, whereas the penile urethra is long and mobile. Its length varies with the length of the penis, stretching up to 40% of its original length during erection [19]. The muscle layer is made up of smooth muscle cells and is surrounded by the outer circular and inner longitudinal muscle layers. Various cell types play an important role in maintaining the functionality of the urethra, as remodeling of the extracellular matrix (ECM) after the injury occurs largely through a coordinated interaction between smooth muscle cells, fibroblasts, and macrophages. Mechanical signals appear to play an important role in maintaining cell phenotype and expression of ECM components [20].

## 3. Sources of Differentiated Cells

A variety of cell sources have been used to engineer urethral tissue constructs. In general, these sources fall into two categories: differentiated cells and stem cells. Figure 1 provides an overview of the various sources that researchers have used or propose to use for urethral tissue engineering. Differentiated cells are cells in the final stage of differentiation, although they may undergo phenotypic changes under certain conditions (e.g., in vitro culturing). Differentiated cells for urethral reconstruction mainly comprise mature cells obtained from the penile tissue, bladder, and oral cavity, including epithelial, fibroblasts, epidermal, mesothelial cells, endothelial cells, and smooth muscle cells. The main types of differentiated cells that have been used in urethral tissue engineering are described in detail in the following subsections and summarized in Table 1.

### 3.1. Urinary Mucosal Keratinocytes

With the primary goal of restoring impaired barrier function, a number of studies have focused on the keratinocytes of the urethral mucosa. In a pioneering study by Romagnoli and coworkers, autologous urethral keratinocytes were used to produce cell sheets that restored normal urothelium and urinary function in boys with hypospadias [21]. The feasibility of using autologous urethral keratinocytes was also demonstrated in a preclinical study by Wang and coworkers, in which the authors seeded amniotic membrane scaffolds with urethral epithelial cells to repair urethral defects in rabbits [22]. Urothelial cells (UC) from bladder biopsies have also been shown to be potentially suitable for urethral tissue engineering [23]. Preclinical experiments in rabbits showed that scaffolds seeded with bladder UCs were able to support epithelial integrity, stratification, and continuity with normal urothelium [24]. Despite successful results with biopsy-derived UCs, these protocols are invasive because of the surgical procedure and trauma to the bladder and urethra. To address this issue, bladder and urethral epithelial cells obtained by bladder irrigation have been increasingly considered. These cells are capable of forming cell colonies with typical epithelial growth morphology that are positive for pan-cytokeratin [25]. Although the cells require a feeder layer for the establishment of primary cultures, propagation without feeder cells could be performed up to 14 passages [26]. Using UCs obtained by bladder irrigation, Fossum and coworkers successfully restored urethral function in patients with severe hypospadias [27]. Preclinical studies in rabbits have shown that these cells have the potential to form a stratified urothelium for the repair of long urethral defects [28]. Overall, these studies have demonstrated that bladder irrigation is a viable and reliable source of UCs and is more likely to be accepted by patients in the clinical setting than bladder biopsy. A limitation is that autologous mucosal keratinocytes cannot be obtained from patients with chronic inflammatory conditions of the urinary tract, e.g., lichen sclerosus.

### 3.2. Oral Mucosal Keratinocytes and Fibroblasts

Keratinocytes of the oral mucosa normally reside in a moist physiological environment similar to that of the urinary tract and can differentiate into uroepithelium in a suitable microenvironment. Oral keratinocytes can be obtained from a mucosal biopsy after enzymatic digestion of the epidermis, which has been mechanically separated from the dermis. The main advantage of using oral keratinocytes is that they are easily accessible and can be biopsied under local anesthesia. A study comparing biopsy specimens from the urethra and oral mucosa found that both urethral and oral mucosal tissues from the same donor can retain their stemness after primary culture and cell expansion [29]. Lingual keratinocytes have also been shown to be an effective cell source for urethral regeneration [30]. Mukocell is a commercial tissue-engineered graft comprising autologous oral epithelial cells on a collagen matrix, which has been developed to avoid the complications associated with oral mucosa harvest [31]. These grafts have shown promising clinical results in the treatment of patients with urethral strictures [32,33,34], suggesting that commercial tissue-engineered products may be a feasible alternative for urinary tract reconstruction in the future.

A potential limitation of oral keratinocytes is their low proliferative capacity and clonogenicity, which hinders their large-scale expansion in vitro that may be required for clinical use. Keratinocytes can normally undergo only a few passages under very stringent conditions as they are prone to apoptosis at low density and differentiation and senescence at the confluence. To address these problems, an increasing number of researchers have proposed a co-culture approach in which keratinocytes and fibroblasts are seeded onto scaffolds. Fibroblasts support keratinocyte adhesion and survival by producing a number of growth factors [35] and also promote terminal urothelial differentiation [36]. Preclinical studies have shown that oral keratinocytes and fibroblasts can successfully repair long urethral defects in a canine model [37]. Bhargava and coworkers have shown in clinical studies that tissue-engineered grafts containing autologous oral epithelium and fibroblasts enhance stratified urothelium formation and urethral function [38,39]. Overall, the results support the co-culture of autologous oral keratinocytes and fibroblasts as a viable approach for urethral regeneration.

### 3.3. Epidermal Keratinocytes

Epidermal keratinocytes can be easily isolated from the foreskin by a minimally invasive approach and expanded in sufficient numbers. Epidermal cells are capable of forming a thick epithelium resistant to a moist environment, which is critical in the urethra. Fu and coworkers used decellularized bladder submucosa seeded with foreskin epidermal cells to repair urethral defects in rabbits [40]. Although urethral continuity was successfully restored, microscopic examination revealed that the graft had a traditional epidermal arrangement rather than urethral transitional epithelium. Because biopsy of the foreskin can lead to complications and deformities at the harvest site, biopsies from other hairless skin sites may be useful. In a preclinical study, Rogovaya and colleagues used skin keratinocytes from rabbit ears for the successful regeneration of urothelium free of hair follicles [41]. However, the feasibility of using epidermal cells from sources other than the foreskin in humans remains to be explored.

### 3.4. Mesothelial Cells

Omentum-derived mesothelial cells represent a promising cell source for regenerative medicine applications, as studies have shown that these cells exhibit phenotypic plasticity [42]. In urethral reconstruction, mesothelial cells isolated from omentum biopsies have shown promise in preclinical studies. Gu and coworkers successfully repaired urethral defects in a rabbit model using acellular bladder matrices seeded with autologous mesothelial cells [43]. In another study by the same group, the authors fabricated granulation tissue tubes that were lined with mesothelial cells prior to implantation. In addition to the new urothelial formation, the cell-seeded grafts showed successful regeneration of the smooth muscle layer and gradually remodeled to a normal urethral architecture [44]. While omentum-derived mesothelial cells may possess some advantages over urinary tract-derived cells, long-term cultures of mesothelial cells appear to be challenging due to early senescence, warranting further studies with this cell type.

### 3.5. Endothelial Cells

All organs contain endothelial cells (ECs), which play an important function in lining arteries, veins, and capillaries that remove waste products from distal tissues, transport immune cells, and supply oxygen and nutrients [45]. Vascularization is currently considered a critical factor for the successful regeneration of the urethra. Thus, ECs are emerging as an important cell type in urethral tissue engineering due to their key role in promoting angiogenesis. In this direction, Imbeault and colleagues developed a tubular urethral graft by seeding human umbilical vein ECs on fibroblast sheets that were subsequently tubularized and seeded with UCs [46]. Using a mouse implantation model, they demonstrated that the endothelialized grafts significantly improved early vascularization and minimized necrosis of the grafted cells. In another study, Heller and colleagues isolated human dermal microvascular ECs from the foreskin, which were used to develop a prevascularized buccal mucosal substitute that could be used for the repair of urethral defects [47]. In addition, studies have shown that the administration of endothelial cells has a significant impact on the restoration of erectile function in the reconstruction of corporal tissue [48,49]. Although endothelial cells have been shown to play a crucial role in promoting angiogenesis, obtaining primary ECs remains a challenge, mainly because of the great functional heterogeneity depending on their location and the presence of contaminating cells [50]. These limitations could be addressed by differentiating ECs from progenitor cells, such as endothelial colony-forming cells (ECFCs), which can efficiently differentiate into mature ECs and promote vascular formation in vitro and in vivo [51].

### 3.6. Smooth Muscle Cells

Smooth muscle cells (SMC) in urethral reconstruction are primarily derived from the bladder and corpus spongiosum. As the most important cell type in the muscle layer, they are critical for improving elasticity and preventing lumen collapse [52]. In addition, SMCs may help support the epithelial–mesenchymal interactions required for normal maturation of the urothelium [53]. Feng and coworkers investigated the efficacy of using corporal smooth muscle cells and lingual keratinocytes seeded onto a porcine acellular corpus cavernosum matrix and showed that tissue-engineered grafts promoted a stratified epithelial layer and organized muscle fiber bundles [54]. Another preclinical study demonstrated that SMC grafts successfully regenerated urethral epithelium and smooth muscle layer in rabbits [55]. A clinical study by Raya-Rivera and coworkers reported successful reconstruction of the urethra with autologous epithelial and bladder SMCs [56]. Overall, regeneration of the muscle layer appears to be as important as epithelial regeneration in urethral repair, as SMCs play a key role in maintaining mechanical properties, supporting epithelial maturation, and promoting vascularization. Therefore, further research efforts should focus on methods to improve the contractile properties of the cells before implantation.

**Table 1 ijms-23-14074-t001:** Summary of main properties of differentiated cells used in urethral tissue engineering.

Cell Type	Source	Advantages	Limitations	Refs.
Mucosal keratinocytes	Urethra	-Accurate urethral phenotype	-Trauma to the urethra -Limited supply and expansion capacity	[21,22]
	Bladder (biopsy)	-Successful preclinical results -Minimally invasive	-Trauma to the bladder -Invasive procedure	[23,24]
	Bladder (washings)	-Ease of harvest -Readily accepted by patients -Successful clinical results	-Limited supply -Requires feeder cells to establish culture -Unavailable in certain patient groups	[25,26,27,28]
	Oral mucosa	-Ease of harvest -Phenotype adapted to wet environment -Successful clinical results	-Limited supply -Poor proliferative capacity	[29,30,31,32,33,34]
Fibroblasts	Oral dermis	-Support survival and adhesion of keratinocytes -Produce growth factors	-Limited supply	[35,36,37,38]
	Skin	-Minimally invasive harvest	-Not extensively characterized	[39]
Epidermal keratinocytes	Foreskin/skin	-Great proliferative capacity -Adapted to wet environment	-Failure to develop transitional epithelium -Biopsy may leave a scar -Unavailable in circumcised patients	[40,41]
Mesothelial cells	Omentum	-Successful preclinical results -Phenotypic plasticity	-Limited supply and expansion capacity	[43,44]
Endothelial Cells (ECs)	Blood vessels	-Promote angiogenesis	-Phenotypic and physiologic variability -Few studies in relation to urethral repair	[46,47,48,49]
Smooth muscle cells (SMCs)	Bladder/ corpus spongiosum	-Improve graft mechanical properties -Promote angiogenesis and epithelial maturation	-Trauma to the bladder/urethra -Invasive procedure	[52,54,55,56]

## 4. Sources of Stem Cells

Although autologous differentiated cells have been suggested to have some potential for urethral regeneration, the invasiveness of the collection procedure, the low proliferation rate, and the adverse effects on diseased donors, such as in cancer or inflammatory diseases, remain serious drawbacks to contend with before a major clinical breakthrough is possible. Consequently, interest has shifted to stem cells, which have a high expansion capacity, can differentiate or transdifferentiate into all relevant phenotypes, and, not unimportantly, can modulate regenerative processes through paracrine activity. There are many types of stem cells, but, for the purposes of this review, only embryonic, induced pluripotent, and mesenchymal stem cells will be discussed in more detail. The major sources and characteristics of stem cells used in urethral regeneration are summarized in Table 2.

### 4.1. Embryonic Stem Cells (ESCs)

The cells in the inner cell mass of the blastocysts can be maintained in tissue culture and multiply infinitely as pluripotent stem cells that are able to differentiate into cells from any of the three germ layers, the endoderm, mesoderm, or ectoderm. Urothelial differentiation from ESCs was first described by Oottamasanthien and coworkers using a mouse model [57]. In vitro experiments later demonstrated the significant role of retinoic acid in inducing the urothelial phenotype [58]. Following these achievements, a method to differentiate the human urothelium was reported [59], and other urethra wall-relevant cell types, such as the smooth muscle [60] and endothelial cells [61] were obtained as well. Along with an unabating effort to deliver a product that conforms to the therapeutic criteria by, e.g., introducing xeno-free culture models with extracellular matrices, biosynthetic surfaces, or chemically defined media, there is a good prospect that the concerns regarding ethics, histocompatibility, and the teratocarcinoma formation [62] will be overcome and the ESCs will eventually see successful use for urethral repair.

### 4.2. Induced Pluripotent Stem Cells (iPSCs)

Because iPSCs can be individually tailored by reprogramming the patient’s own somatic cells, researchers have been provided with a powerful tool that is not burdened by the serious obstacles associated with the ESCs, such as the ethical and histocompatibility problems [63]. The iPSCs have been eagerly explored by several groups in the field of urethral reconstruction, and it has been consistently concluded that it is feasible to induce and drive them along the urothelial specification pathway [59,64,65]. Nevertheless, there are still some unresolved issues, including low reprogramming and differentiation efficiency and tumorigenicity, that need to be resolved before the full clinical potential of iPSCs can be realized [66].

### 4.3. Mesenchymal Stem Cells (MSCs)

In general, these cells have a mesodermal tri-lineage differentiation potential into the bone, cartilage, and fat, and exhibit a limited capacity to transdifferentiate. They are important for the homeostasis of different stem cell niches through their paracrine and immunomodulatory activities and direct intercellular contact, which is why they are widely spread throughout the body. Depending on the origin, many different variants can be found within this category, and most of them have been investigated for urethral reconstruction. The review will further focus on the following six types, including bone marrow-derived stem cells (BMSCs), adipose-derived stem cells (ASCs), urine-derived stem cells (USCs), hair follicle-associated stem cells (HFSCs), amniotic fluid-derived mesenchymal stem cells (AF-MSCs), and umbilical cord-derived mesenchymal stem cells (UCB-MSCs). The last type represents a distinct heterogeneous category since many reports do not rigorously differentiate between the cells originating from the dermal mesenchymal compartment and the hair shaft epithelial stem cells and progenitors.

Regarding BMSC, Tian and coworkers have shown that using co-culture and conditioned medium protocols, these cells were able to differentiate into both urothelial- and SMC-like cells [67]. Another report documented the benefits of BMSC in healing wound-associated inflammation in a rat model [68]. Despite the clear potential, there are issues primarily associated with the collection, such as low cell yields or procedure-related morbidities, that may be of concern, especially when considering broader practical use [69].

The ASCs, on the other hand, can be isolated from subcutaneous adipose tissue with up to 500-fold higher yield without increasing the risk of complications [70]. Over 400,000 liposuction procedures are performed each year, with up to 3 L of lipoaspirate discarded after each procedure [71]. It is possible to collect up to 6 billion ASCs after a single passage by collecting and processing all discarded tissue [72,73], and this easy accessibility made ASCs highly sought after for regenerative purposes. Studies have shown that ASCs can be differentiated into urothelial-like cells under appropriate microenvironmental conditions, and a co-culture format with UCs and/or conditioned medium appears especially effective [74,75,76]. There is additional evidence that the ASCs can undergo epithelial trans-differentiation in a three-dimensional (3D) arrangement using scaffolds, either decellularized or synthetic [77,78]. Since the ASCs have also been demonstrated to yield upon induction a smooth muscle phenotype [79,80,81], they appear to be an appealing source even for highly intricate urethral engineering applications. Aside from differentiation, ASCs can secrete a variety of biologically active molecules, such as growth factors, cytokines, and chemokines, promoting an anti-inflammatory environment, angiogenesis, and wound healing. Interestingly, ASCs are not only a source of cells for urethral reconstruction but they can be induced to synthesize ECM scaffolds [82]. It should be underscored that such scaffolds are autologous, and this presents a significant benefit when it comes to biocompatibility issues and surgical success rates [83]. Taken together, despite many beneficial features, the procedures required for the in vitro differentiation of ASCs into UCs or SMCs are complex and do not offer a distinct advantage over the autologous UCs or SMCs [5].

USCs appear to be another particularly attractive stem cell for urethral reconstruction. They have a high potential for self-renewal and differentiation, anti-inflammatory and anti-fibrotic properties and, in contrast to the archetypal BMSCs and ASCs, can be obtained non-invasively and, thus, at low cost [84,85]. In addition, the donor’s age, gender, or health status, with the exception of urinary tract infection and anuria, does not seem to play a role, and the USC cultures have been successfully established even from donors with end-stage bladder cancer [86]. The USCs display a high level of stemness and proliferative capacity so that a single 24 h urine collection can yield over three passages of more than 1 × 10^8^ cells, which is enough for nearly any application [87,88,89]. Apart from being an almost limitless source, the USCs were demonstrated to differentiate into functional UCs [86], SMCs [86], and endothelial cells [90] to support urothelial mucosa, muscle wall, and blood vessels, respectively. These properties were further explored in a 3D setting, where Wu and coworkers were able to produce a tissue construct that was reminiscent of the native urothelial and SMC layers [91]. In summary, the USCs represent an excellent source for urological tissue engineering applications. They are innate to the urinary tract and, thus, can survive exposure to urine similarly to healthy UCs. However, the viability in urine may be compromised unless appropriate environmental control is in place, which is why more research on the storage and transfer methods is needed.

Regarding the hair follicles, there is limited data, yet HFSCs have been shown to be able to support a urothelial-like phenotype in tissue culture or urothelium-like stratified constructs on the scaffold construct [92,93]. In light of the previously documented plasticity of cells in the hair follicle dermal papilla and sheet [94], it is plausible that it is these MSCs that respond with trans-differentiation, but proof remains to be provided. From a practical point of view, HFSCs are definitely attractive because their source is easily accessible. On the other hand, the difficulties related to the limited isolation yield and the general lack of deeper understanding make these cells only a second choice for any urethral tissue engineering project.

**Table 2 ijms-23-14074-t002:** Summary of main properties of stem cells used in urethral tissue engineering.

Cell Type	Source	Advantages	Limitations	Refs.
Embryonic stem cells (ESCs)	Human embryos	-Can be differentiated to any cell type in the urethra -Suitable for in vitro models	-Ethical issues -Malignant potential -Time-consuming differentiation process	[57,58,59,60,61]
Induced pluripotent stem cells (iPSCs)	Reprogrammed cells from adult tissues	-Can be differentiated to any cell type in the urethra -Suitable for in vitro models -No ethical issues -Can be used for patient-specific grafts	-Low reprogramming and differentiation efficiency -Malignant potential -Time-consuming differentiation process	[58,64,65]
Mesenchymal stem cells (MSCs)	Bone marrow (BMSCs)	-Extensively characterized -Promote neovascularization	-Invasive procedure -Some donor morbidity -Low yield	[67,68]
	Adipose tissue (ASCs)	-Extensively characterized -Easy to harvest -Highly abundant -Low donor morbidity -Broad paracrine effects	-Inhomogeneous cell population	[74,75,76,77,78,79]
	Urine (USCs)	-Non-invasive harvest -Great proliferative capacity	-Compromised viability after long exposure to urine -Not extensively characterized	[86,90,91]
	Hair follicles (HFSCs)	-Minimally-invasive harvest -Great proliferative capacity	-Limited supply -Not extensively characterized -More studies are needed in relation to urethral repair	[92,93]
	Amniotic fluid (AF-MSCs)	-Great proliferative capacity -Differentiation potential toward urothelial lineage	-Limited supply -Not extensively characterized	[95]
	Umbilical cord blood (UCB-MSCs)	-Mostly similar to BMSCs -Easy to harvest -Great proliferative capacity	-More studies are needed in relation to urethral repair	[96]

The other two MSCs, including those found in the amniotic fluid and umbilical cord, share many common features because they originate in extraembryonic gestational tissue. In general, both AF -MSCs and UCB-MSCs provide an optimal balance between quality and ethics, as they are abundant and pose minimal ethical and legal issues. Importantly, they have been successfully induced in vitro in urothelial mimicking phenotypes [95,96] and further data demonstrate their potential to repopulate scaffold constructs in vivo [97]. Another advantage of these MSCs is that they represent an allogeneic cell source that could be used “off the shelf” after the identification of cell lines with potentially higher efficacy. Although current clinical results suggest that allogeneic MSCs are safe and provide clinically meaningful efficacy [98], treatment with allogeneic cells still poses the risk of developing long-term alloreactivity, which requires better long-term monitoring of any unexpected adverse reactions in clinical trials [99]. As with MSCs as a whole, further research is required before pregnancy tissue-derived MSCs are accepted as a clinically viable modality.

## 5. Cell Preconditioning Approaches

A number of preconditioning and engineering strategies have been developed in recent years with the goal of maintaining cell viability, improving cell survival, enhancing cell maturation and differentiation, and promoting angiogenesis. Some of the most prominent are dealt with in more detail below.

### 5.1. Biomimetic Microenvironmental Approaches

It is desirable for any urethral engineering application that the biopsy samples are best possible protected en route from the donor. With this in mind, researchers have recently demonstrated that the transport of oral tissue samples using thermoreversible gelation polymers provides an optimal environment for preserving the viability of oral epithelial cells intended for tissue-engineered grafts [100]. Scaffolds that mimic the natural physical environment of cells are also critical to support cell survival, attachment and synthesis of ECM, prevent apoptosis, and facilitate cell migration. In this context, nanofibrous scaffolds fabricated by electrospinning of biopolymers provide an architecture that mimics the ECM and enables the incorporation of relevant biomolecules during the fabrication process [101,102]. Wang and coworkers have demonstrated the successful reconstruction of the urethra in a rabbit model using poly-L-lactic acid (PLLA) scaffolds seeded with ASCs [78]. Electrospun silk fibroin scaffolds have also shown excellent results in urethral reconstruction [103]. Interestingly, three-layer electrospun scaffolds that mimic the architecture of the native urethra seeded with oral fibroblasts and keratinocytes not only supported better cell attachment and proliferation but also possessed the mechanical properties of natural tissue [104].

Natural scaffolds obtained by the decellularization of tissues may also provide an inductive environment to support cell attachment and maturation prior to implantation. In a preclinical study, type I collagen cell carriers (CCC) with stratified multilayered autologous urethral epithelium were used to perform urethroplasty in minipigs [105]. The implanted grafts successfully integrated into the host with concomitant development of junctional complexes and differentiation, suggesting that the collagen matrix may improve graft stability. Although the results of using scaffolds that mimic the native architecture of the urethra are promising, it must be emphasized that most preclinical studies have been performed in animal models with transient urethral defects. However, these animal models are very different from patients in whom the pathogenic condition for urethral stricture is scarring of the corpus spongiosum leading to fibrosis.

### 5.2. Surface Modification and Cell Seeding Technology

Despite the encouraging results obtained so far with unmodified synthetic and natural scaffolds in the field of urethral regeneration, new preconditioning strategies, such as the incorporation of bioactive molecules and the optimization of cell seeding technology are being considered. One such strategy is to functionalize the surface of synthetic scaffolds with naturally occurring ECM molecules or peptide sequences to enhance cell adhesion. Using this approach, Uchida and coworkers modified the surface of polycarbonate-urethane-urea scaffolds, known to have mechanical properties similar to those of bladder tissue, with fibronectin and gelatin, which improved the affinity of urothelial and bladder smooth muscle cells [106]. The porosity of a scaffold is primarily responsible for the infiltration of stromal cells. However, even in scaffolds with high porosity, spontaneous infiltration takes a long time. Therefore, methods have been developed in which dynamic culture increases cell infiltration of scaffolds [107]. Compared with static seeding techniques, agitation and centrifugation result in better infiltration as far as the stromal cells are concerned, however, when it comes to the other cell types relevant to urethral regeneration more work needs to be done [108,109].

### 5.3. Scaffold-Free Approaches

Cell sheet engineering is a technique widely used in regenerative medicine, including urethral reconstruction. Cells are grown in culture surfaces containing a temperature-responsive polymer, the poly-N-isopropylacrylamide (PIPAAm). At 37 °C, the PIPAAm forms a dense membrane that supports cell attachment and proliferation [110]. When the temperature drops below its critical temperature (32 °C), the polymer swells and becomes hydrophilic, leading to spontaneous detachment of the cell layer [111]. This approach allows harvesting of the cells and deposited ECM without proteolytic treatment, maintaining cell adhesion molecules and important growth factors bound to the ECM. Zhou and coworkers used a dog model to demonstrate the utility of the cell layer technology in urethral reconstruction. They created tissue constructs from ASCs, oral mucosal epithelial cells, and fibroblasts that were successfully used for structural and functional regeneration of the urethra [29]. Compared to conventional scaffold materials, cell sheets exhibit higher cell concentration, more uniform cell distribution, higher cell viability, and no immune system activation caused by scaffold materials. Cell sheets in clinical use today are derived from autologous cells, which reduces the risk of immune rejection, but the current cost required to produce patient-derived cell sheets severely limits their widespread use. Off-the-shelf allografts may be an option in the future; however, further research is needed to determine how to manage risks associated with immune rejection and/or transmission of infection.

### 5.4. Bioprinting

Bioprinting is a relatively new technology, but it already has shown great potential for producing complex cell-laden constructs that can be tailored to specific needs [112]. A unique advantage is that it can be used to create different cellular structures to mimic the complexity of natural tissues, which is not possible with conventional scaffold-based technology [113]. Using 3D printing technologies, Zhang and coworkers fabricated a structure mimicking the structural and mechanical properties of the rabbit urethra [114]. A 3D-printed spiral tubular scaffold served as a support for two cell-loaded hydrogel layers in the outer and inner surfaces, containing SMCs and UCs, respectively. Although the maturation of the cells was not investigated, this study provided the first proof of concept that bioprinting is a promising approach to assembling the different layers of the urethra in predefined spatial patterns. In another study, Pi and coworkers developed a multichannel coaxial extrusion technique for printing tubular structures with multiple circumferential layers. With this technology, and using a sodium alginate and gelatin methacrylate (GelMA) blend bioink loaded with human UCs and SMCs, they printed tubular structures that mimicked urethral tissue [115]. Their results show that bioprinting not only allows for high structural fidelity but also that the cells retain the ability to proliferate and differentiate. These results support the use of bioprinting in urethral tissue engineering, but issues, such as biomaterial selection, fine-tuning of printing parameters, crosslinking time, and mechanical properties need to be addressed to optimize the functional performance of the constructs.

### 5.5. Bioreactors

Bioreactors are systems that provide a more physiologically relevant environment for cultures compared to traditional static conditions and allow for organ modeling in vitro. By regulating pH, temperature, oxygen partial pressure, cell perfusion, and external mechanical stimuli, these systems support tissue development by providing the biochemical and physical regulatory signals required for cell proliferation, differentiation, and ECM production [116]. Simultaneous application of biophysical and biochemical stimulation signals in the bioreactors results in synergistic responses that are expected to significantly improve the functional properties of the cells [117]. Wang and coworkers investigated the feasibility of dynamic mechanical stimulation to promote smooth myogenic differentiation of ASCs seeded on polyglycolic acid (PGA) [118]. After one week of static culture, tubular cell-PGA constructs were induced by 5-azacytidine (5-aza) and stimulated in a pulsatile flow bioreactor for five weeks. Histological examination revealed that the urethral-shaped constructs contained smooth muscle-like cells and well-oriented collagen fibers. Similarly, Yang and colleagues were able to employ the pulsed-flow conditions to achieve the formation of a fully developed multilayer UC epithelial layer within the tubular collagen scaffold [119]. In line with this study, Versteegden and coworkers developed a system to mimic the urine flow stress on the human urethra, demonstrating that mechanical stimulation is critical for maintaining a tight epithelial layer [120]. While significant progress has been made in the design, construction, and application of bioreactors for urethral tissue engineering, most bioreactors are currently dedicated devices with low-volume output. The optimal culture conditions for different cell types on different scaffolds need to be further optimized.

### 5.6. Addition of Bioactive Factors

Recent studies have shown that growth factors can be incorporated into the tissue-engineered constructs to meet cell growth and maturation requirements, and also to support the development of a functional vasculature that is critical for graft survival [29,121,122]. For example, Loai and coworkers, when experimenting with a bladder acellular matrix scaffold coated with VEGF, achieved the formation of new blood vessels as well as urothelial and smooth muscle layers in the constructs engrafted in rats and pigs [123]. In an attempt to provide for a more sustained signaling, a recombinant VEGF protein containing the collagen-binding domain (CBD-VEGF) was devised, and indeed it was documented as superior to simple VEGF in terms of neovascularization in a dog model [124].

Before urethral regeneration can become embraced as a reliable option, there is the critical issue of stricture recurrence due to tissue fibrosis, which needs to be resolved. The TGF-β1 is believed to be the culprit, thus targeting its receptor and/or signaling pathways appears of key importance, as exemplified by targeting the canonical Wnt regulatory pathway [125,126]. In this context, Zhang and coworkers introduced the Wnt pathway inhibitor ICG-001 into electrospun scaffolds [127]. Urethrography results showed patent urethra in all rabbits of the ICG-001 group, in contrast to the control group where the urethral strictures and fistulas were frequent. Li and coworkers produced a tissue-engineered urethral graft using oral keratinocytes and fibroblasts transfected with TGF-β1 siRNA [128]. The result was a decrease in collagen deposition, effectively inhibiting fibrosis. Overall, the results of these studies indicate that targeted delivery or inhibition of growth factors in the tissue-engineered grafts is a valid approach that may translate into improved in vivo performance.

## 6. Concluding Remarks and Perspectives

Due to a number of shortcomings in the practical application of autologous grafts and cell-free scaffolds, tissue engineering with cells remains an important research direction for large urethral defects. In this review, the main cell types and current techniques in the field of urethra tissue engineering have been recapitulated. Although a number of scaffold materials, cell types, and tissue engineering techniques have been developed, there is still no consensus on the choice of specific options.

The differentiated cells isolated from patient biopsies and bladder irrigation are currently the mainstay source, however, the focus is shifting towards the somatic stem cells, in particular the ASCs and USCs, due to their wide differentiation and trans-differentiation ability and an extensive biological regulatory potential. While autologous cells remain the preferred cell source, future efforts should focus on evaluating allogeneic cells that could help increase the availability and efficacy of urethral regeneration therapies. In addition to cells, the scaffolds are central to regenerative efforts since they can alter cellular responses in a decisive manner. Consequently, a major impetus is being dedicated to the development of scaffolds that better mimic the ECM in terms of microstructure and surface chemistry, and especially the advent of the 3D bioprinting technology opens up solutions that were hitherto inconceivable. In perspective, further research is needed to create microenvironmental conditions conducive to the different urethral cell phenotypes, which requires the development of dedicated biomaterials and manufacturing processes.

## Figures and Tables

**Figure 1 ijms-23-14074-f001:**
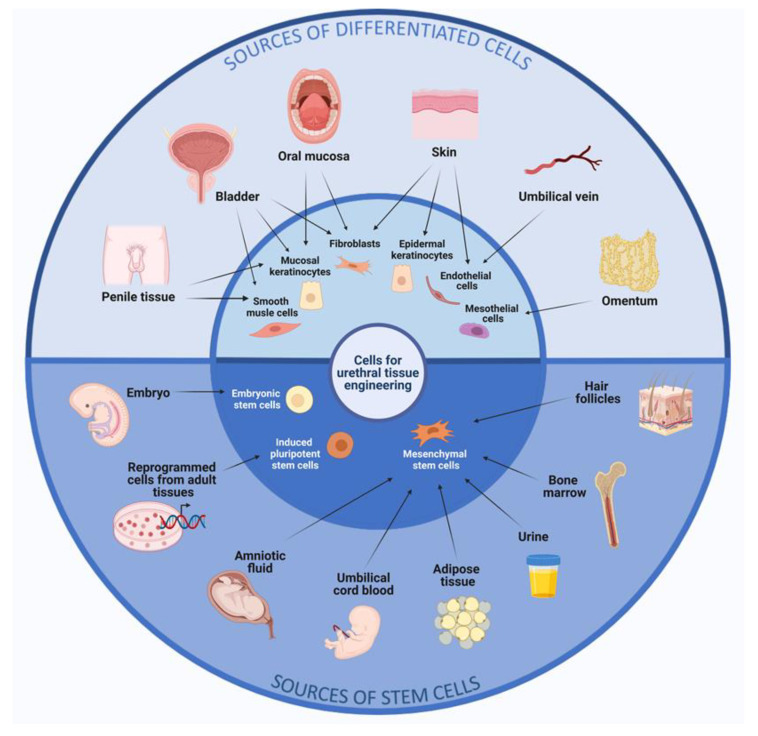
Schematic overview of the different tissue sources for cells relevant to urethral tissue engineering. The sources are divided into two categories: differentiated cells and stem cells.

## Data Availability

Not applicable.

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
