# Peer review of "Sources, Selection, and Microenvironmental Preconditioning of Cells for Urethral Tissue Engineering"

_ijms, 2022, doi:10.3390/ijms232214074_

Round 1

Reviewer 1 Report

See attached word file

Reviewer 2 Report

Sources, selection, and microenvironmental preconditioning of cells for urethral tissue engineering

IJMS-2010-864

Abstract:

Adequate and to the point.

Introduction:

Good summary of the clinical situation leading to this review. Article aim was clearly stated.

Sections:

Figure 1 gives a good overview.

Table 1, a bladder biopsy performed by cystoscopy is not that invasive. We usually say »minimally invasive» same on line 118, page 5. Well explained on lines 128-9

Line 184, we would challenge this statement that « Although obtaining mesothelial cells from the omentum may be easier than obtaining biopsies from the urinary tract» To obtain cells from the omentum, we need to perform an intra-abdominal biopsy, usually by laparoscopy. How is it easier?

In 3.5 are SMC really SMCs? As noted on line 202, they lose their contractile phenotype. They act more like fibroblasts. Their most important role is probably to be organ-specific, which will help obtaining a better urothelial coverage. Described by Bouhout et al. in World Journal of Urology: 2016, Jan, 34(1), 121-30

Good section 4 on stem cells. 4.2 Great argument, iPSCs are not where some people think they are. Nobody has convincingly showed a nice urothelium using these cells.

Discussion and perspectives:

Limitations are presented.

Even if some elements are not present in the review, it offers a good perspective on most of the aspects involved in urethral tissue engineering.

Round 2

Reviewer 1 Report

The manuscript has been massively improved in this revision round. Thank you for inserting the importantance of vasculature in the review. One last (minor) improvement could be the source of the endothelial cells. These cells can be differentiated from MSCs (from different origin), another way to isolate these cells is from circulating ECFC, as described e.g. in this review https://www.frontiersin.org/articles/10.3389/fmed.2018.00295/full

Author Response

We thank the reviewer for the positive feedback. The use of endothelial progenitors as source for endothelial cells, as well as the suggested reference, have been included in the new version of the manuscript